# DAVIS: HIGH-QUALITY AUDIO-VISUAL SEPARATION WITH GENERATIVE DIFFUSION MODELS

## ABSTRACT

We propose DAVIS, a **D**iffusion model-based **A**udio-**VI**usal **S**eparation framework that solves the audio-visual sound source separation task through a generative manner. While existing discriminative methods that perform mask regression have made remarkable progress in this field, they face limitations in capturing the complex data distribution required for high-quality separation of sounds from diverse categories. In contrast, DAVIS leverages a generative diffusion model and a Separation U-Net to synthesize separated magnitudes starting from Gaussian noises, conditioned on both the audio mixture and the visual footage. With its generative objective, DAVIS is better suited to achieving the goal of high-quality sound separation across diverse categories. We compare DAVIS to existing state-of-the-art discriminative audio-visual separation methods on the domain-specific MUSIC dataset and the open-domain AVE dataset, and results show that DAVIS outperforms other methods in separation quality, demonstrating the advantages of our framework for tackling the audio-visual source separation task.

## 1 INTRODUCTION

Visually-guided sound source separation, also referred to as audio-visual separation, is a pivotal task for assessing a machine perception system's ability to integrate multisensory signals. The primary goal is to isolate individual sounds from a complex audio mixture by utilizing visual cues about the objects that are producing the sounds, e.g., separate the "barking" sound from the mixture by querying the "dog" object. To achieve human-like intelligence, an effective separation model should be capable of handling a *diverse* range of sounds and produce *high-quality* separations that can deliver a realistic auditory experience.

The community has dedicated significant efforts to this task, and existing methods (Zhao et al., 2018; Gao & Grauman, 2019; Gan et al., 2020; Chatterjee et al., 2021; Tian et al., 2021; Dong et al., 2022; Zhu & Rahtu, 2022; Chen et al., 2023) have made extensive attempts to tackle this problem, such as developing more powerful separation frameworks (Zhao et al., 2018; Gao & Grauman, 2019; Chatterjee et al., 2021; Chen et al., 2023), proposing more effective training pipelines (Tian et al., 2021), and incorporating additional visual cues (Gan et al., 2020) to enhance the separation performance. For optimization, these approaches usually take mask regression (Zhao et al., 2018) or spectrogram reconstruction (Owens & Efros, 2018) as training objectives.

While these methods have shown promising separation performance in specific domains, such as musical instrument sounds, they are not yet satisfactory in dealing with open-domain sounds where background noise and off-screen sounds are prevalent. These sounds produce complicated mosaic of time and frequency patterns, posing significant challenges in achieving high-quality separation. Thus, a natural question arises: *is there an effective approach to model these complex audio data distribution and produce high-quality separated sounds?*

We answer the question by introducing a generative framework for the audio-visual separation. A new class of generative models called denoising diffusion probabilistic models (DDPMs) (Ho et al., 2020; Nichol & Dhariwal, 2021; Song et al., 2020a), also known as diffusion models, has emerged recently and demonstrated remarkable abilities in generating diverse and high-quality images (Dhariwal & Nichol, 2021) and audio (Kong et al., 2020). The impressive capabilities of generative diffusion models in capturing complex data distributions inspire us to explore their potential for enhancing audio-visual separation. Unlike discriminative modeling, we believe that generative diffusion models

can effectively approximate more intricate data distributions, allowing us to handle open-domain time and frequency patterns and lead to superior separation results.

To this end, we present DAVIS, a novel framework for audio-visual separation that is built upon a generative diffusion model. Unlike typical discriminative methods that predict a mask representing the separated sound from the input mixture, DAVIS approaches the separation task as a conditional generation process. Specifically, our method incorporates a T-step diffusion and reverse process Ho et al. (2020); Dhariwal & Nichol (2021); Nichol & Dhariwal (2021): during the training stage, Gaussian noise controlled by a variance schedule (Jabri et al., 2022) is added to the unmixed sound at each diffusion step. In the reverse process, our method initiates from a standard Gaussian distribution, and an effective Separation U-Net is proposed to estimate the noise added at each diffusion step, iteratively generating the separated magnitude with guidance from the mixture and visual footage. The Separation U-Net comprises an encoder-decoder structure with enabled skip connections. To capture both local and long-range time-frequency patterns, we introduce a Convolution-Attention (CA) block consisting of a ResNet block for capturing local patterns, an efficient Time-Frequency Attention block to learn long-range time-frequency correlation, and a Time Attention block to enhance the time dependencies. Furthermore, to enhance audio-visual association learning, we devise a Feature Interaction module to facilitate interactions between audio and visual features and inject visual cues into the separation.

Experiments on the MUSIC (Zhao et al., 2018) and AVE (Tian et al., 2018) datasets demonstrate that DAVIS outperforms the state-of-the-art methods in terms of separation quality. Our contributions are summarized as follows:

- We are the first study, to the best of our knowledge, to approach the audio-visual separation task as a conditional generation process and solve it using a diffusion model.
- We design a Separation U-Net, which incorporates CA blocks and a Feature Interaction module to capture the audio-visual association effectively.
- Our framework surpasses previous methods on both specific and open-domain sound datasets, highlighting the benefits of solving audio-visual separation through a generative approach.

## 2 RELATED WORK

**Audio-Visual Sound Source Separation.** In this section, our focus is on modern audio-visual sound source separation approaches while acknowledging the prolonged research efforts dedicated to sound source separation in signal processing. Recent deep learning-based audio-visual sound source separation methods have been applied to a variety of audio categories, including speech signals (Ephrat et al., 2018; Owens & Efros, 2018; Afouras et al., 2020; Michelsanti et al., 2021), musical instrument sounds (Zhao et al., 2018; Gan et al., 2020; Tian et al., 2021; Gao & Grauman, 2019; Zhao et al., 2019; Chatterjee et al., 2021; Tan et al., 2023), and universal sound sources (Gao et al., 2018; Mittal et al., 2022; Tzinis et al., 2020; 2022; Chatterjee et al., 2022; Zhu & Rahtu, 2022; Dong et al., 2023; Chen et al., 2023). These methods typically employ a learning regime that involves mixing two audio streams from different videos to provide supervised training signals. A sound separation network, often implemented as a U-Net, is then used for mask regression (Zhao et al., 2018) conditioned on associated visual features. In recent years, research in this area has focused on both domain-specific and open-domain sound source separation (Tzinis et al., 2020; Mittal et al., 2022; Zhu & Rahtu, 2022; Dong et al., 2023; Chen et al., 2023). However, existing methods often require additional information, such as text queries (Dong et al., 2023), motion cues (Mittal et al., 2022; Zhu & Rahtu, 2022), or class labels (Chen et al., 2023), to achieve good performance. In this paper, we propose a novel generative audio-visual separation approach that outperforms existing methods in separating both specific and open-domain sound sources.

**Diffusion Models.** Diffusion models (Ho et al., 2020; Song et al., 2020b; Song & Ermon, 2019) fall under the category of deep generative models that start with a sample in a random distribution and gradually restore the data sample through a denoising process. Recently, diffusion models have exhibited remarkable performance across various domains, including computer vision (Dhariwal & Nichol, 2021; Avrahami et al., 2022; Ramesh et al., 2022; Gu et al., 2022; Nichol et al., 2021; Ho et al., 2022; Singer et al., 2022; Ruiz et al., 2022; Saharia et al., 2022), natural language processing (Austin et al., 2021; Gong et al., 2022; Li et al., 2022; Chen et al., 2022b), and audio applications (Kong et al., 2020; Popov et al., 2021; Lee & Han, 2021; Chen et al., 2022b; 2020; Huang et al., 2022;

Scheibler et al., 2023). While diffusion models have been successfully employed for single-modality generation, their potential for audio-visual tasks remains largely unexplored. For instance, only recently has MM-diffusion (Ruan et al., 2022) proposed simultaneous generation of videos and audio. Furthermore, there has been a growing interest in utilizing diffusion models for discriminative tasks. Some pioneer works have explored the application of diffusion models to image segmentation (Amit et al., 2021; Baranchuk et al., 2021; Brempong et al., 2022) and object detection (Chen et al., 2022a). However, despite significant interest in this direction, there have been no prior successful attempts to apply generative diffusion models to audio-visual scene understanding, which has notably lagged behind the progress in visual perception tasks. To the best of our knowledge, this paper presents the first work that adopts a diffusion model to learn audio-visual associations for audio-visual sound source separation.

## 3 METHOD

In this section, we introduce DAVIS, our novel diffusion model-based audio-visual separation framework designed for achieving high-quality separation results. We begin by providing a brief recap of the preliminary knowledge of diffusion models in Sec. 3.1. Next, we present our proposed Separation U-Net architecture, which effectively captures the audio-visual association through the generation process in Sec. 3.3. Finally, we discuss the training and inference pipelines in Sec. 3.4.

### 3.1 PRELIMINARIES

We introduce the concept of diffusion models, which serves to illustrate the pipeline of our framework. A diffusion model consists of a forward and a reverse process. The forward process is defined as a Markov chain that gradually adds noise to the data sample $x_0$ according to a variance schedule $\beta_1, ..., \beta_T$:

$$q(x_{1:T}|x_0) = \prod_{i=1}^{T} q(x_t|x_{t-1}), \tag{1}$$

$$q(x_t|x_{t-1}) = \mathcal{N}(x_t|\sqrt{\alpha_t}x_{t-1}, \beta_t \mathbf{I}), \tag{2}$$

where $\alpha_t = 1 - \beta_t$ and $\bar{\alpha}_t = \prod_{s=0}^{t} \alpha_s$. Note that the variance schedule is also fixed during the reverse process. If the total number of $T$ goes to infinity, the diffusion process will finally lead to pure noise, *i.e.*, the distribution of $p(x_T)$ is $\mathcal{N}(x_t; \mathbf{0}, \mathbf{I})$ with only Gaussian noise.

The reverse process aims to recover samples from Gaussian distribution by removing the noise gradually, which is a Markov chain parameterized by $\theta$:

$$p_\theta(x_{0:T}) = p(x_T) \prod_{t=1}^{T} p_\theta(x_{t-1}|x_t), \tag{3}$$

where at each iteration, the noise $\epsilon$ added in the forward process is estimated as:

$$p_\theta(x_{t-1}|x_t) = \mathcal{N}(x_{t-1}; \boldsymbol{\mu_\theta}(x_t, t), \boldsymbol{\Sigma_\theta}(x_t, t)). \tag{4}$$

Note that we set the variances $\boldsymbol{\Sigma_\theta}(x_t, t) = \tilde{\beta}_t \mathbf{I}$ to untrained constants, and $\boldsymbol{\mu_\theta}(x_t, t)$ is typically implemented as neural networks. Unlike vanilla diffusion models, the output of our separation framework depends on both audio mixture and visual information. To adapt the reverse process into a conditional one, we include the conditional context $\boldsymbol{c}$ as additional network inputs, which modifies Eq. 4 as follows:

$$p_\theta(x_{t-1}|x_t) = \mathcal{N}(x_{t-1}; \boldsymbol{\mu_\theta}(x_t, t, \boldsymbol{c}), \tilde{\beta}_t \mathbf{I}), \quad \text{where } \tilde{\beta}_t := \frac{1 - \bar{\alpha}_{t-1}}{1 - \bar{\alpha}_t} \beta_t. \tag{5}$$

To train the network, we follow (Ho et al., 2020) to adopt a simplified training objective:

$$L_{simple}(\theta) = \mathbb{E}_{t,x_0,\epsilon}[||\epsilon - \epsilon_\theta(\sqrt{\bar{\alpha}_t}x_0 + \sqrt{1 - \bar{\alpha}_t}\epsilon, \boldsymbol{c}, t)||], \tag{6}$$

where $\epsilon_\theta$ represents a function approximator used to predict $\epsilon$ (the noise added at each iteration in the forward process according to Eq. 2), while $t$ denotes a uniformly sampled value ranging from 1 to $T$.

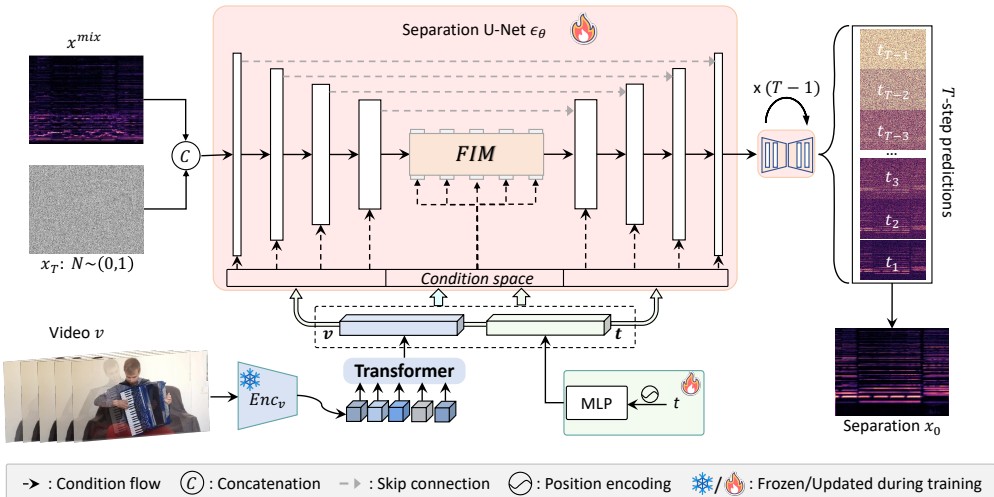

Figure 1: **Overview of the DAVIS framework.** Our objective is to synthesize the separated sound $x_0$ by leveraging an audio mixture $x^{mix}$ and the synchronized visual stream $v$, while taking into account the diffusion timestep $t$. Firstly, we sample a latent variable $x_T$ from a standard distribution. Next, we encode the frames $v = \{I_j\}_{j=1}^K$ and the timestep $t$ into the embedding space. We use a transformer to produce the discriminate visual feature vector $\boldsymbol{v}$. The features $\boldsymbol{v}$ and $\boldsymbol{t}$ serve as conditions in the Separation U-Net $\epsilon_\theta$, which performs iterative denoising on $x_T$ to produce the separated sound $x_0$. Specifically, $\boldsymbol{t}$ is passed to all the modules within $\epsilon_\theta$, while $\boldsymbol{v}$ is only used in the Feature Interaction Module (Sec. 3.3) for audio-visual association learning.

## 3.2 Task Setup and Method Overview

Given an unlabeled video clip $V$, we can extract an audio-visual pair $(a, v)$, where $a$ and $v$ are the audio and visual stream, respectively. In real-world scenarios, the audio stream can be a mixture of $N$ individual sound sources, denoted as $a = \sum_{i=1}^N s_i$, where each source $s_i$ can be of various categories. Meanwhile, the visual stream $v$ is typically a synchronized video of $K$ images, denoted as $v = \{I_j\}_{j=1}^K$, where the $I_j$ are the individual frames of the video. The primary goal of the visually-guided sound source separation task is to utilize visual cues from $v$ to effectively separate $a$ into its constituent sources $s_i$, for $i \in \{1, 2, ..., N\}$. Since no labels are provided to distinguish the sound sources $s_i$, prior works (Zhao et al., 2018; Tian et al., 2021; Huang et al., 2023) have commonly used a "mix and separate" strategy, which involves mixing audio streams from two different videos and manually create the mixture: $a^{mix} = a^{(1)} + a^{(2)}$. Furthermore, the time series $a$ is usually transformed into magnitude spectrogram by short-time Fourier transform (STFT): $x = \mathbf{STFT}(a) \in \mathbb{R}^{T \times F}$, allowing for manipulations in the 2D-like Time-Frequency domain, where $F$ and $T$ are the numbers of frequency bins and time frames, respectively. Consequently, the goal of training is to learn a separation network capable of mapping $\boldsymbol{f} : (\boldsymbol{x^{mix}}, \boldsymbol{v^{(1)}}) \rightarrow \boldsymbol{x^{(1)}}$. For simplicity, we will omit the video index notation in the subsequent sections [1].

In contrast to discriminative approaches that perform the mapping through regression, our proposed DAVIS framework is built on a diffusion model with a T-step diffusion and reverse process. The diffusion process is determined by a fixed variance schedule as described in Eq. (1) and Eq. (2), which gradually adds noises to the magnitude spectrogram $x_0$ and converts it to latent $x_T$. As depicted in Fig. 1, the reverse process (according to Eq. (3) and Eq. (5)) of DAVIS is specified by our proposed separation network $\epsilon_\theta$. This reverse process iteratively denoises a latent variable $x_T$, which is sampled from a uniform distribution, to obtain a separated magnitude conditioned on the magnitude of the input sound mixture $x^{mix}$ and the visual stream $v$. Consequently, the objective of the separation network $\epsilon_\theta$ is to predict the noise $\epsilon$ added at each diffusion timestep during the forward process.

The challenges of solving audio-visual separation task are threefold: (C1) learning informative audio features that can represent different sound components; (C2) learning discriminative visual features

---

[1]In this paper, superscripts denote video indices, while subscripts refer to diffusion timesteps.

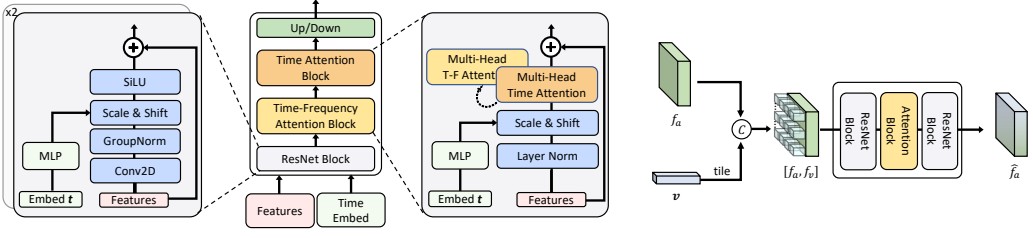

(a) Architecture of CA Block      (b) Audio-Visual Feature Interaction Module (FIM)

Figure 2: **Illustrations on CA Block and Feature Interaction Module.** (a) Our CA block operates by taking audio feature maps and a time embedding $t$ as inputs. Each sub-block, except the up/down sampling layer, is conditioned on $t$. It consists of two groups of convolutions within each ResNet block to capture local time-frequency patterns, while the Attention blocks capture long-range dependencies along the time and frequency dimensions. (b) The Feature Interaction Module functions by replicating and concatenating $v$ with $f_a$. Two identical ResNet blocks and an attention block, as described in (a), are used to process the concatenated features.

that summarize the sound-related video content; and (C3) capturing precise audio-visual association to perform separation. In the following sections, we will introduce our designs for tackling these three challenges sequentially.

### 3.3 PROPOSED DAVIS FRAMEWORK

Previous works often use U-Net-like (Ronneberger et al., 2015) architectures for separation network designs, which is attributed to its effectiveness in capturing multi-level feature representations and producing separated magnitudes of the same size as inputs. Exploiting the grid-like nature of magnitude spectrograms, existing methods employ convolution-based U-Nets and concatenate audio and visual features directly at the bottleneck to incorporate visual cues. While these approaches achieve good separation performance, we argue that they may be inadequate in addressing the three challenges (Sec. 3.2) of real-world sound separation: (C1) Similar frequency patterns can occur in temporally distant frames, and distinct frequency patterns can mix within a single time frame. Such occurrences necessitate the network to capture both local patterns and long-range dependencies across time and frequency dimensions, which pure convolution (Zhao et al., 2018; Gao & Grauman, 2019) may fall short. (C2) Real-world videos often have mismatched visual and audio content. Learning visual condition from frame features (Tian et al., 2021; Dong et al., 2023) without considering the possible unrelated audio-visual content can hence lead to less discriminative visual cues. (C3) Capturing accurate audio-visual associations is crucial, but directly concatenating visual and audio embeddings at the bottleneck (Gao & Grauman, 2019) lacks the ability to foster further interactions between audio and visual modalities. To address these challenges, we propose a novel Separation U-Net that incorporates Convolution-Attention blocks to learn both local and global time-frequency associations, introduce a simple yet effective temporal transformer to aggregate the frame features, and devise an audio-visual feature interaction module to enhance association learning by enabling interactions between audio and visual modalities.

**Encoder/Decoder Designs.** Our proposed Separation U-Net architecture comprises an encoder and a decoder, connected by an audio-visual feature interaction module. Both the encoder and decoder consist of five Convolution-Attention (CA) Blocks, and skip connections are used to facilitate information flow. Initially, we concatenate the latent variable $x_T$ with the mixture $x^{mix}$ along the channel dimension and use a 1x1 convolution to project it to the feature space. As depicted in Fig. 2, each CA block consists of a ResNet block, an efficient Time-Frequency Attention block, and a Time Attention block. Following this, a downsample layer (or upsample layer for the decoder) with a scale factor of 2 is used. Specifically, we construct the ResNet block using WeightStandardized 2D convolution (Qian et al., 2020) along with GroupNormalization (Wu & He, 2018) and SiLU activation (Elfwing et al., 2018). To incorporate the time embedding $t$ as a conditioning factor, we employ an MLP to generate $t$-dependent scaling and shifting vectors for feature-wise affine transformation (Dumoulin et al., 2018) before the activation layer. To reduce computational complexity, we adopt an efficient form of attention mechanism (Shen et al., 2021) for the Time-Frequency Attention block. A Time Attention block is then appended to enhance long-range time dependencies. For implementation, we adopt the

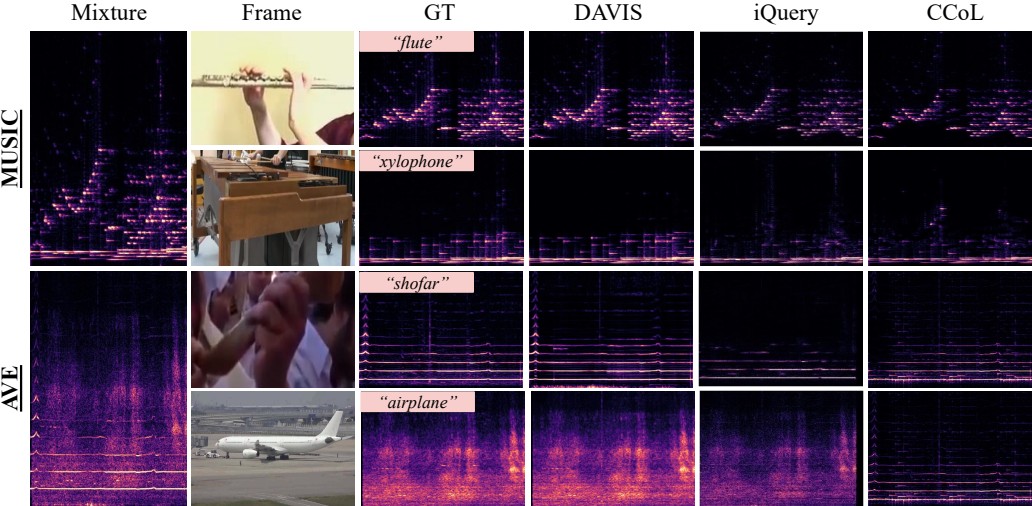

Figure 3: Visualizations of audio-visual separation results on the MUSIC (top) and AVE (bottom) datasets. Two sounds are mixed (mixture), and referenced frames are provided to guide the separation. We show the comparison between ground truth and DAVIS/iQuery/CCoL's predictions.

design proposed by Wang et al. (2023), which includes Pre-Layer Normalization and Multi-Head Attention along the time dimension within the residual connection. The downscale and upscale layers are implemented using 2D convolutions with a stride of 2. As a result, we can obtain audio feature maps $\boldsymbol{f_a} \in \mathbb{R}^{C \times \frac{T}{32} \times \frac{F}{32}}$ at the bottleneck, where $C$ represents the number of channels. Additionally, we include a 1x1 convolution to convert the decoder output into magnitude.

**Timestep Embedding.** In a diffusion model, the timestep embedding serves to inform the model about the current position of the input within the Markov chain. As shown in Fig. 1, diffusion time $t$ is specified by the Transformer sinusoidal positional encoding (Vaswani et al., 2017) and further transformed by an MLP, which will be passed to each CA block as a timestep condition.

**Visual Condition Aggregation.** Not all frames in a video will be attributable to the synchronized audio. To account for unaligned visual content, we incorporate a shallow transformer to effectively learn the visual condition. Specifically, we extract frame features $\{I_j\}_{j=1}^K$ from the visual stream $v = \{I_j\}_{j=1}^K$ using a pre-trained ResNet-18 (He et al., 2016) visual backbone $\mathbf{Enc_v}$, where $I_j \in \mathbb{R}^C$. We apply a self-attention temporal transformer $\phi(\cdot)$ to aggregate raw visual frame features, resulting in $\{\hat{I}_j\}_{j=1}^K = \phi(\{I_j\}_{j=1}^K)$. For the transformer design, we empirically found that a shallow transformer with three encoder layers and one decoder layer works well. The global visual embedding $v$ is then computed by averaging the temporal dimension of $\{\hat{I}_j\}_{j=1}^K$.

**Audio-Visual Feature Interaction Module.** The key to successful audio-visual separation lies in effectively utilizing visual information to separate visually-indicated sound sources. Therefore, the interaction between audio and visual modalities at the feature level becomes crucial. Existing approaches typically concatenate audio and visual features at the bottleneck (Gao & Grauman, 2019; Chatterjee et al., 2021) and pass them to the decoder for further fusion. In this paper, we propose an audio-visual feature interaction module to enhance this capability. We spatially tile $v$ to match the shape of $\boldsymbol{f_a}$, resulting in visual feature maps $\boldsymbol{f_v}$. Subsequently, the audio and visual feature maps are concatenated along channel dimension and fed into the feature interaction module (FIM): $\hat{\boldsymbol{f}}_a := \mathbf{FIM}([\boldsymbol{f_a}, \boldsymbol{f_v}])$, where $\hat{\boldsymbol{f}}_a \in \mathbb{R}^{C \times \frac{T}{32} \times \frac{F}{32}}$. The details of the **FIM** module are illustrated in Fig. 2(b), encompassing ResNet blocks and a Time-Frequency Attention block that facilitates the establishment of audio-visual associations in both local and global regions.

## 3.4 TRAINING AND INFERENCE

Given the sampled audio-visual pairs from the dataset, we first adopt the "mix and separate" strategy and compute the magnitudes $x^{(1)}, x^{(2)}, x^{mix}$ with STFT.

| Methods | LF | Output | MUSIC (Zhao et al., 2018) | | | AVE (Tian et al., 2018) | | |
|---|---|---|---|---|---|---|---|---|
| | | | SDR↑ | SIR↑ | SAR* | SDR↑ | SIR↑ | SAR* |
| Mixture | - | - | 0.31 | 0.31 | 149.90 | 0.30 | 0.30 | 149.78 |
| SoP (Zhao et al., 2018) | ✓ | Mask | 3.42 | 4.98 | - | 0.46 | 4.17 | 12.08 |
| CoSep (Gao & Grauman, 2019) | ✗ | Mask | 2.04 | 6.21 | - | -1.33 | 2.54 | 5.77 |
| CCoL (Tian et al., 2021) | ✓ | Mask | 7.18 | 12.55 | 11.09 | 1.77 | 3.25 | 22.52 |
| AMnet (Zhu & Rahtu, 2022) | ✓ | Mask | 6.16 | 8.66 | 12.86 | 2.85 | 5.20 | 12.14 |
| CLIPSep (Dong et al., 2023) | ✓ | Mask | 3.44 | 4.73 | 18.00 | 2.19 | 3.51 | 16.26 |
| iQuery† (Chen et al., 2023) | ✗ | Mask | 10.89 | 14.95 | 14.81 | 3.88 | 6.82 | 12.86 |
| DAVIS | ✓ | Mag. | 11.18 | 18.06 | 14.63 | 3.77 | 7.95 | 10.34 |

Table 1: Comparison of our method to other discriminative audio-visual separation methods on the MUSIC and AVE test sets. "LF" denotes "Label-Free", meaning that only visual frames are needed for separation. "Output" shows the difference between our method (magnitude synthesis) and the others (mask regression). Note that iQuery† requires class labels both in training and inference. We report SDR, SIR, and SAR metrics, and highlight our results in gray . SAR* is to measure the artifacts and might not fully reflect separation quality (discussed in Sec. 4.2).

**Data Scaling:** To align with the frequency decomposition of the human auditory system, we apply a logarithmic transformation to the magnitude spectrogram, converting it to a log-frequency scale. Additionally, we ensure consistent scaling of the log-frequency magnitudes by multiplying them with a scale factor $\sigma$ and clipping the values to fall within the range [0, 1].

The visual frames are encoded to embeddings $\boldsymbol{v}^{(1)}, \boldsymbol{v}^{(2)}$. Taking video (1) as an example, we sample $\epsilon$ from a standard Gaussian distribution and $t$ from the set $\{1, ..., T\}$. Then, we input $x_t^{(1)}, x^{mix}, \boldsymbol{v}^{(1)}, t$ to the Separation U-Net $\epsilon_\theta$ and optimize the network by taking a gradient step on Eq. (6). In practice, we use both video (1) and (2) for optimization, therefore the final loss term is formulated as $\mathcal{L} = \mathcal{L}_{simple}^{(1)}(\theta) + \mathcal{L}_{simple}^{(2)}(\theta)$. The training objective enforces the model to reconstruct the individual sound by utilizing the audio mixture and visual information. As a result, our model will not "hallucinate" content as generation tasks but predict results that are parts of the mixture.

Our inference process starts from a sampled latent variable $x_T$, and takes the mixture $x^{mix}$ and visual frame embedding $\boldsymbol{v}$ as conditions to produce the separated magnitude $x_0$ through T iterations. Finally, the output is rescaled to the original value range. The detailed pseudo codes of training and inference are provided in the Appendix D.

## 4 EXPERIMENTS

### 4.1 EXPERIMENTAL SETUP

**Datasets.** Our model demonstrates the ability to handle both specific and open-domain sound separation problems. To evaluate our approach, we use MUSIC (Zhao et al., 2018) and AVE (Tian et al., 2018) datasets, which cover musical instruments and open-domain sounds. The evaluation settings are described in detail below:

- **MUSIC**: We evaluate our proposed method on the widely-used MUSIC (Zhao et al., 2018) dataset, which includes 11 musical instrument categories: accordion, acoustic guitar, cello, clarinet, erhu, flute, saxophone, trumpet, tuba, violin, and xylophone. All the videos are clean solo and the sounding instruments are usually visible. We follow Tian et al. (2021) and use the same train/val/test splits, resulting in 468/26/26 videos across various instrument categories.

- **AVE**: In addition to the MUSIC dataset, we also evaluate our method on the Audio-Visual Event (AVE) dataset (Tian et al., 2018). This dataset contains 4143 10-second videos, including 28 diverse sound categories, such as *Church Bell*, *Barking*, and *Frying (Food)*, among others. The AVE dataset presents greater challenges as the audio in these videos may not span the entire duration and can be noisy, including off-screen sounds (e.g., human speech) and background noise. We conduct training and evaluation on this demanding dataset using the original train/val/test splits, consisting of 3339/402/402 videos, respectively.

**Baselines.** To the best of our knowledge, we are the first to adopt a generative model for the audio-visual source separation task. Thus, we compare DAVIS against the following discriminative methods: (i) *Sound of Pixels* (SoP) (Zhao et al., 2018) that learns ratio mask predictions with a 1-frame-based

| Fusion | SDR↑ | SIR↑ | SAR↑ |
|---|---|---|---|
| Concat | 10.85 | 17.62 | 15.52 |
| FIM (Point-wise) | 11.06 | 17.37 | 15.44 |
| FIM (Local) | 11.56 | 17.02 | 16.28 |
| FIM (Global) | 11.23 | 17.56 | 15.84 |
| FIM (Local&Global) | 11.88 | 17.52 | 16.12 |

| Block | SDR↑ | SIR↑ | SAR↑ |
|---|---|---|---|
| {R, R, R} | 9.03 | 14.05 | 13.20 |
| {R, R, T} | 11.78 | 17.91 | 15.44 |
| {R, R, TF} | 11.50 | 18.01 | 15.21 |
| {R, TF, T} | 11.88 | 17.52 | 16.12 |

| Sampling step | SDR↑ | SIR↑ | SAR↑ |
|---|---|---|---|
| Step=10 | 11.82 | 17.47 | 16.03 |
| Step=15 | 11.82 | 17.51 | 15.93 |
| Step=25 | 11.88 | 17.52 | 16.12 |
| Step=50 | 11.84 | 17.58 | 15.83 |

Table 2: Ablation studies. **Left:** Ablation on Feature Interaction Module. **Middle:** Ablation on CA block design. R, TF and T denote ResNet, Time-Frequency Attention and Time Attention blocks, respectively. **Right:** Number of sampling steps.

model, (ii) *Co-Separation* (CoSep) (Gao & Grauman, 2019) that takes a single visual object as the condition to perform mask regression, (iii) *Cyclic Co-Learn* (CCoL) (Tian et al., 2021) which jointly trains the model with sounding object visual grounding and visually-guided sound source separation tasks, (iv) *AMnet* (Zhu & Rahtu, 2022) which is a two-stage framework modeling both appearance and motion, (v) *CLIPSep* (Dong et al., 2023) that leverages the powerful CLIP (Radford et al., 2021) model to learn text-queried sound separation with noisy unlabeled videos, and (vi) *iQuery* (Chen et al., 2023) that adapts the maskformer architecture for audio-visual separation and achieves the current state-of-the-art (SOTA) results. We use the entire image for CoSep and CCoL on the AVE dataset because it lacks bounding box annotations for detected objects. These methods can still work well on the AVE dataset because the videos are usually clean and contain only one object. Therefore, the global visual features of the entire image and the detected object are similar. For all the comparative methods, we use the authors' publicly available code.

**Evaluation Metrics.** To quantitatively evaluate the audio-visual sound source separation performances, we use the standard metrics (Zhao et al., 2018; Tian et al., 2021; Gao & Grauman, 2019), namely: Signal-to-Distortion Ratio (SDR), Signal-to-Interference Ratio (SIR), and Signal-to-Artifact Ratio (SAR). We adopt the widely-used mir eval library (Raffel et al., 2014) to report the standard metrics. Note that SDR and SIR evaluate the accuracy of source separation, whereas SAR specifically measures the absence of artifacts (Gao & Grauman, 2019). Consequently, SAR can be high even if the separation performance is poor in terms of accurately separating the sources.

## 4.2 COMPARISONS WITH STATE-OF-THE-ART

To evaluate the effectiveness of our method, we present separation results by comparing DAVIS with state-of-the-art approaches on the MUSIC and AVE datasets, as depicted in Tab. 1. Our results highlight the advantages of utilizing generative modeling for audio-visual separation. DAVIS consistently outperforms previous approaches across various evaluation categories, achieving similar SDR results and up to a 3.1 dB and a 1.1 dB improvement on the SIR scale for the MUSIC and AVE dataset, surpassing the performance of the next best approach iQuery. These results clearly demonstrate the versatility of our method across diverse datasets with varying visual and audio contexts. Among the competing approaches, we observe that some of them yield higher SAR results than ours but have lower SDR/SIR values. We argue that high SAR values do not necessarily imply effectiveness, as they can arise from poor separation. It is worth noting that a comparison between the mixture spectrogram and the ground truth unmixed spectrogram can surprisingly yield high SAR values (first row in Tab. 1). In this context, we believe that our method significantly improves separation performance compared to others. In Fig. 3, we visually compare our separation results to the iQuery/CCoL baselines. Our visualizations demonstrate that DAVIS achieves higher separation quality, as evidenced by the closer resemblance of our separated magnitude spectrograms to the ground truth. Moreover, the successful handling of diverse time patterns in the provided examples highlights the importance of incorporating attention mechanisms in our Separation U-Net.

## 4.3 EXPERIMENTAL ANALYSIS

We conduct ablations on the MUSIC validation set to examine the different components of DAVIS. **Block Design.** We validate the effectiveness of our proposed CA block (shown in Fig. 2 (a)) by designing the following baselines: (a) Using three consecutive ResNet blocks within the CA block, which only captures local time-frequency patterns; (b) Replacing the last ResNet block with a Time Attention block; (c) Replacing the last ResNet block with a Time-Frequency Attention block; and (d) replacing the last two ResNet blocks with Time-Frequency and Time attention blocks to enhance the ability to capture long-range dependencies. The results presented in Tab. 2 demonstrate the

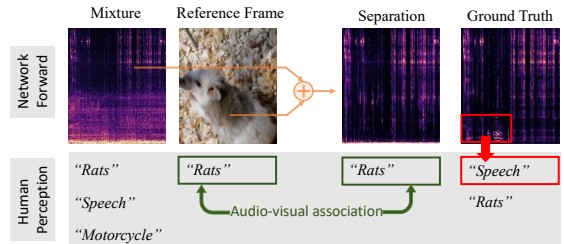

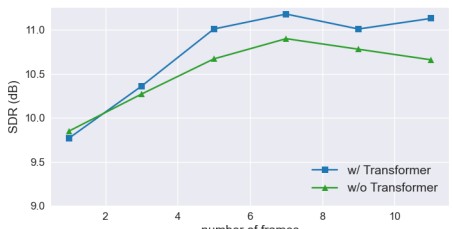

Figure 4: An visualization example showing that our DAVIS model can capture accurate audio-visual association to perform visually-guided separation.

Figure 5: Ablation on varying the number of frames to validate the effect of our proposed temporal transformer.

significance of capturing capturing both local patterns and long-range dependencies across time and frequency dimensions. In our implementation, the attention blocks have fewer parameters than the ResNet block, indicating that the improvement is not coming from increasing network capacity.

**Aggregated Visual Condition.** To study the effect of the visual condition, we vary the number of sampled frames and compare models with and without the temporal transformer, as shown in Fig. 5. The results demonstrate that increasing the number of frames achieves a more informative visual condition and boosts separation quality. When the number of frames is large, noisy information may be incorporated, resulting in a performance drop. We show that adopting a temporal transformer can mitigate this issue and lead to a consistent separation performance.

**Audio-Visual Feature Interaction.** We conduct an ablation study on the Feature Interaction Module (FIM) to validate the importance of effective audio-visual association learning for this task. Specifically, we explore different ways of feature interaction: (a) direct concatenation of visual and audio features, (b) three-layer MLP for point-wise fusion, (c) three ResNet blocks, (d) three attention blocks, and (e) a combination of ResNet and attention blocks. The results in Tab. 2 show that a naive concatenation of audio and visual features performs significantly worse while allowing for further interaction between them improves the results. Among all the designs, our proposed FIM module achieves the best results by considering both local and global contexts.

**Sampling Step Analysis.** We investigate the impact of varying the number of sampling steps in Tab. 2. From the results, we select the step value as 25, while observing that satisfactory results are obtained even with step = 10. This suggests that further acceleration is possible if faster inference speed is prioritized. Excitingly, early exploration (Song et al., 2023) indicates that diffusion models can perform single-step inference from any arbitrary timestep to the final step, strengthening this potential for acceleration.

**Learned Audio-Visual Association.** The learned audio-visual associations are essential for successful separation. To demonstrate the accuracy of our model's learned associations, we show an example from the AVE dataset in Fig. 4. In this example, a video clip labeled "Rats" is mixed with another video clip labeled "Motorcycle." However, human perception reveals the presence of an off-screen sound "Speech" occurring in the "Rats" clip, while only the "rat" object is visible in the reference frame. In this scenario, our method successfully separates the "Rats" sound from the complicated mixture while disregarding the "Speech" and "Motorcycle" sounds, thus affirming the accuracy of our learned audio-visual associations and our method's capability to capture complex data distribution.

## 5 CONCLUSION

In this paper, we propose DAVIS, a diffusion model-based audio-visual separation framework designed to address the problem in a generative manner. Unlike approaches relying on discriminative training objectives for regression, our separation framework is built upon a T-step diffusion model, allowing for the iterative synthesis of the separated magnitude spectrogram while conditioning on the visual footage. Leveraging the power of generative modeling, our method effectively handles complex data distributions and achieves high-quality sound separation. Extensive experiments on the MUSIC and AVE datasets validate the efficacy of our framework, demonstrating its effectiveness in separating sounds within specific domains (e.g., music instrument sounds) as well as its ability to generalize to open-domain sound categories.

**Reproducibility Statement.** We follow Tian et al. (2021) to process the MUSIC (Zhao et al., 2018) and AVE (Tian et al., 2018) datasets. The network structure is specified in Fig. 1 and Fig. 2. For each submodule in Fig. 2, we elaborate it in the Sec. 3.3, with clear details and references. The training and inference pipelines are illustrated in pseudo codes in Appendix D, and the hyperparamters are provided in Appendix C.

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

## A  Demo Page

We have included a demo page in order to illustrate our method and showcase the separation results. This page comprises an example of iterative synthesis that allows users to interact. Furthermore, we present several separation examples of DAVIS against iQuery (Chen et al., 2023) on both the MUSIC (Zhao et al., 2018) and AVE (Tian et al., 2018) datasets.

# B    DISCUSSION

## B.1    SIGNIFICANT DIFFERENCE FROM NAIVE CONDITIONAL DIFFUSION MODELS

We would like to clarify that directly extending conditional diffusion models for audio-visual sound separation does not work. This is because conditional diffusion models are not designed to handle the unique challenges of audio-visual separation, such as the need to leverage sound-relevant visual information to help with separating individual sound sources.

To tackle this problem, we proposed a CA block and Feature Interaction Module to model complicated time-frequency patterns in audio spectrograms and capture audio-visual associations. These two modules allow our model to learn more about the relationships between audio and visual signals, which is essential for effective audio-visual sound separation.

In addition, for the diffusion model itself, we noticed the importance of noise scheduling. We found that the commonly used linear scheduler is not effective in our scenario (as shown in Fig. 6). This is because the audio spectrogram is a sparse signal, with high-energy patterns confined to specific regions. The linear schedule leads to predominant noise accumulation during most steps, posing considerable learning challenges. To address this issue, we adopted non-linear noise schedulers (Nichol & Dhariwal, 2021; Jabri et al., 2022) that delicately control the amount of noise to add. This scheduler allows our model to learn more about the underlying audio signal without being overwhelmed by noise.

We believe that our approach is a significant improvement over previous methods for audio-visual sound separation. We have demonstrated its effectiveness on two challenging datasets, MUSIC (Zhao et al., 2018) and AVE (Tian et al., 2018), and we believe that it has the potential to be used in a variety of applications, such as video editing, sound mixing, and hearing aid design.

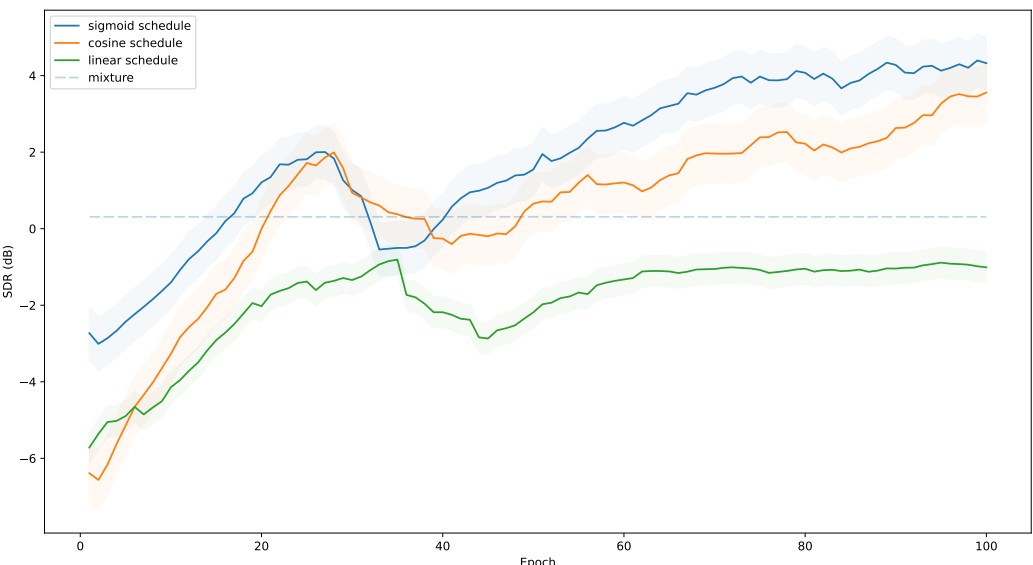

Figure 6: Plot of the SDR results from models trained with different noise schedules. The model trained with the popular linear schedule cannot separate the sounds.

## B.2    DIFFERENCE BETWEEN AUDIO DIFFUSION SEPARATION WORK AND DAVIS

Our work is a new take on the audio-visual source separation problem and is inherently different from the audio diffusion separation approaches. Compared to DiffSep (Scheibler et al., 2023), the most recent audio speech source separation work that employs a diffusion model, our proposed DAVIS framework is different in several aspects:

**1) The diffusion strategies.** While DiffSep (Scheibler et al., 2023) applies the diffusion process in the time domain, our proposed DAVIS framework is operated in the spectrogram domain. Specifically,

DiffSep starts with separated sources and gradually mixes them as more noise is added. Its reverse process starts from the mixture and ends with separated sources. In contrast, our DAVIS framework starts from the separated sound and gradually adds noise to it, transforming it into pure Gaussian noise instead of the mixture. Then the reverse process in our framework starts with the noise, conditioned on the audio mixture and the visual cues, and ends with separated sources.

**2) The framework architecture.** Our framework has a different pipeline from DiffSep (Scheibler et al., 2023): Our method learns audio-visual correlations and guides source separation using visual cues, while DiffSep is an unconditional model that works on audio data only. In terms of architecture, DiffSep uses the same network architecture as conventional diffusion models. Conversely, we propose a Separation U-Net to effectively learn audio-visual associations for the audio-visual separation task.

## C    IMPLEMENTATION DETAILS

In our experimental setup, we down-sample audio signals at 11kHz. For the MUSIC dataset, the video frame rate is set to 8 fps. Each video is approximately 6 seconds and we uniformly select 11 frames per video. As for the AVE dataset, we set the video frame rate to 1 fps (following the setup of Tian et al. (2018)). We use the entire 10-second audio as input and use 10 frames to train the model. During training, the frames are first resized to $256 \times 256$ and then randomly cropped to $224 \times 224$. We set the total diffusion time step $T = 1000$ to train our DAVIS model. During inference, all the frames are directly resized to the desired size without cropping. To accelerate the separation process, we use DDIM (Song et al., 2020a) with a sampling step of 25. The audio waveform is transformed into a spectrogram with a Hann window of size 1022 and a hop length of 256. The obtained magnitude spectrogram is subsequently resampled to $256 \times 256$ to feed into the separation network. We set the number of audio and visual feature channels $C$ as 512 and empirically choose the scale factor $\sigma = 0.15$. Our model is trained with the Adam optimizer, with a learning rate of $10^{-4}$. The training is conducted on a single A6000 GPU for 200 epochs with a batch size of 8.

---

**Algorithm 1** Training

---

1: **Input:** A dataset $D$ that contains audio-visual pairs $\{(a^{(k)}, v^{(k)})\}_{k=1}^{K}$, total diffusion step $T$
2: **Initialize:** randomly initialize Separation U-Net $\epsilon_\theta$ and temporal transformer $\phi(\cdot)$, and load the pre-trained visual encoder $\mathbf{Enc_v}$
3: **repeat**
4:     Sample $(a^{(1)}, v^{(1)})$ and $(a^{(2)}, v^{(2)}) \sim D$
5:     Mix and compute $x^{mix}, x^{(1)}$
6:     Scale $x = \log_e(1 + x) \cdot \sigma$ and clip $x^{mix}, x^{(1)}$ to [0,1]
7:     Encode visual frames $v^{(1)}$ as $\boldsymbol{v^{(1)}} := \phi(\mathbf{Enc_v}(v^{(1)}))$
8:     Sample $\epsilon \sim \mathcal{N}(\mathbf{0}, \mathbf{I})$, and $t \sim \text{Uniform}(1, ..., T)$
9:     Take gradient step on
10:      $\nabla_\theta ||\epsilon - \epsilon_\theta(x_t^{(1)}, x^{mix}, \boldsymbol{v^{(1)}}, t)||, x_t^{(1)} = \sqrt{\bar{\alpha}_t} x^{(1)} + \sqrt{1 - \bar{\alpha}_t}\epsilon$
11: **until** converged

---

## D    TRAIN AND INFERENCE PSEUDO CODE

The complete training procedure for our DAVIS framework is shown in Algorithm 1. Given the sampled audio-visual pairs from the dataset, we first use the "mix and separate" strategy to create the mixture, and compute the magnitudes $x^{(1)}, x^{(2)}, x^{mix}$ using STFT. We then apply a logarithmic transformation to the magnitude spectrogram to convert it to a log-frequency scale. Finally, we ensure consistent scaling of the log-frequency magnitudes by multiplying by a scale factor $\sigma$ and clipping to the range [0, 1].

The visual frames are encoded to embeddings with the pre-trained visual backbone and aggregated by a trainable temporal transformer followed by an averaging operation. This gives us the visual conditions $\boldsymbol{v^{(1)}}, \boldsymbol{v^{(2)}}$. For the training process, taking video (1) as an example, we sample $\epsilon$ from a standard Gaussian distribution and $t$ from the set $\{1, ..., T\}$. Then, we input $x_t^{(1)}, x^{mix}, \boldsymbol{v^{(1)}}, t$ to the Separation U-Net $\epsilon_\theta$ and optimize the network by taking a gradient step on Eq. (6). In

---

**Algorithm 2** Inference

1: **Input:** Audio mixture $a^{mix}$ and the query visual frame $v$, total diffusion step $T$
2: Sample $x_T \sim \mathcal{N}(\mathbf{0}, \mathbf{I})$
3: Compute $x^{mix} := \mathbf{STFT}(a^{mix})$
4: Encode visual frames $v$ as $\boldsymbol{v}^{(1)} := \phi(\mathbf{Enc_v}(v))$
5: **for** $t = T, ..., 1$ **do**
6:     Sample $z \sim \mathcal{N}(\mathbf{0}, \mathbf{I})$ if $t > 1$, else $z = 0$
7:     Compute $x_{t-1}$: $x_{t-1} = \frac{1}{\sqrt{\alpha_t}}(x_t - \frac{1-\alpha_t}{\sqrt{1-\bar{\alpha}_t}}\epsilon_\theta(x_t, x^{mix}, \boldsymbol{v}, t)) + \sqrt{\tilde{\beta}_t}z$
8: **end for**
9: **return** $e^{x_0/\sigma} - 1$

---

practice, we use both video (1) and (2) for optimization, so the final loss term is defined as $\mathcal{L} = \mathcal{L}^{(1)}_{simple}(\theta) + \mathcal{L}^{(2)}_{simple}(\theta)$.

As illustrated in Algorithm 2, our inference process starts from a sampled latent variable $x_T$, and takes the mixture $x^{mix}$ and visual condition $\boldsymbol{v}$ to produce the separated magnitude $x_0$ through T iterations. Finally, the output is rescaled to the original range.

## E MORE QUALITATIVE VISUALIZATIONS

In this section, we provide more visualizations on the MUSIC and AVE datasets against iQuery/CCoL, and an additional example to demonstrate our learned audio-visual association.

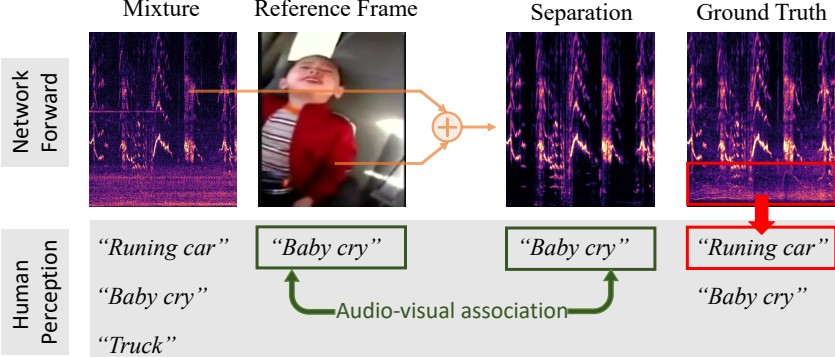

Figure 7: An additional visualization example showing that our DAVIS model can capture accurate audio-visual association to perform visually-guided separation.

**Learned Audio-Visual Association.** In Fig. 7, a video clip labeled "Baby cry" is mixed with another video clip labeled "Truck." However, human perception reveals the presence of an off-screen sound "Running car" occurring in the "Baby cry" clip, while only the "baby" object is visible in the reference frame. In this scenario, our method successfully separates the "Baby cry" sound from the complicated mixture while disregarding the "Running car" and "Truck" sounds, thus affirming the accuracy of our learned audio-visual associations and our method's capability to capture complex data distribution and deal with more than two sounds in the mixture.

**More Visualizations on the MUSIC Dataset.** In Fig. 8, we show audio-visual separation results across different instrument categories and compare our method with iQuery and CCoL. It's clear that DAVIS achieves higher separation quality when dealing with various time and frequency patterns. We encourage readers to visit our demo page and listen to the results for a better comparison.

**More Visualizations on the AVE Dataset.** In Fig. 9, we show audio-visual separation results on more challenging scenarios, where distinct time-frequency patterns and background noise exist. In the first and the seventh rows, background noise exists in the "rats" and "bell" videos (marked in the

red box). In these cases, our method successfully discards the background noise and separates only the sound related to the given visual frame, demonstrating DAVIS's strong audio-visual association learning capability. We encourage readers to visit our demo page and listen to the results for a better comparison.

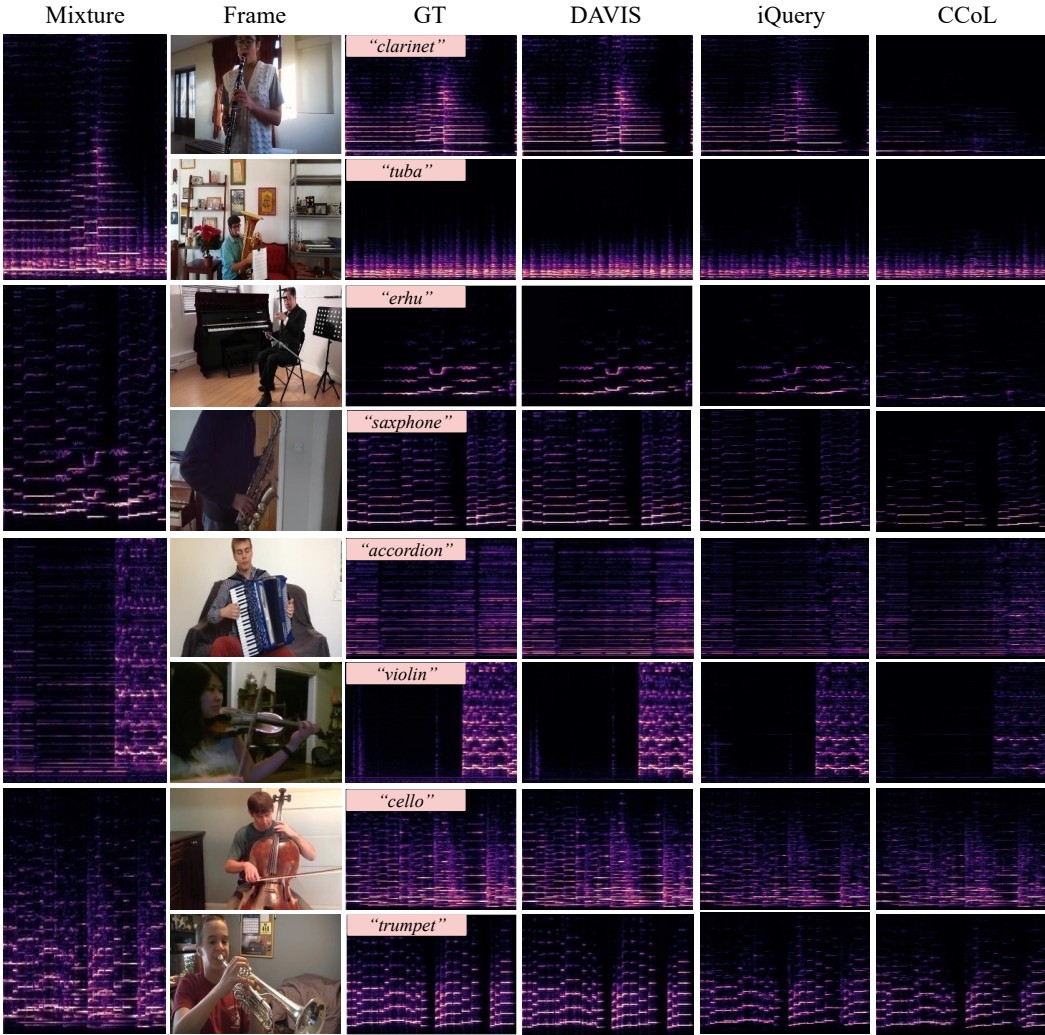

Figure 8: Visualizations of audio-visual separation results on the MUSIC dataset. Two sounds are mixed (mixture), and the referenced frame is provided to guide the separation. We show the comparison between ground truth and DAVIS/iQuery/CCoL's predictions.

**Result on the Natural Sound Mixture.** In Fig. 10, we provide a real-world separation example on a natural video with two instruments. Although our method is trained on artificial sound mixtures (generated by manually mixing two sounds), DAVIS successfully separates the individual "violin" and "guitar" sounds from a YouTube video.

**Dealing with mixtures of more than two sounds.** Although our model is trained on mixtures of two videos, DAVIS can yield good separation results even with mixtures involving more than two sound sources during inference. We verify this claim through examples in Figs. 4 and 7, and AVE Example 1 & 2 on the demo page. The examples show that DAVIS can successfully separate desired sounds through visual conditioning, even in cases with three concurrent sound sources.

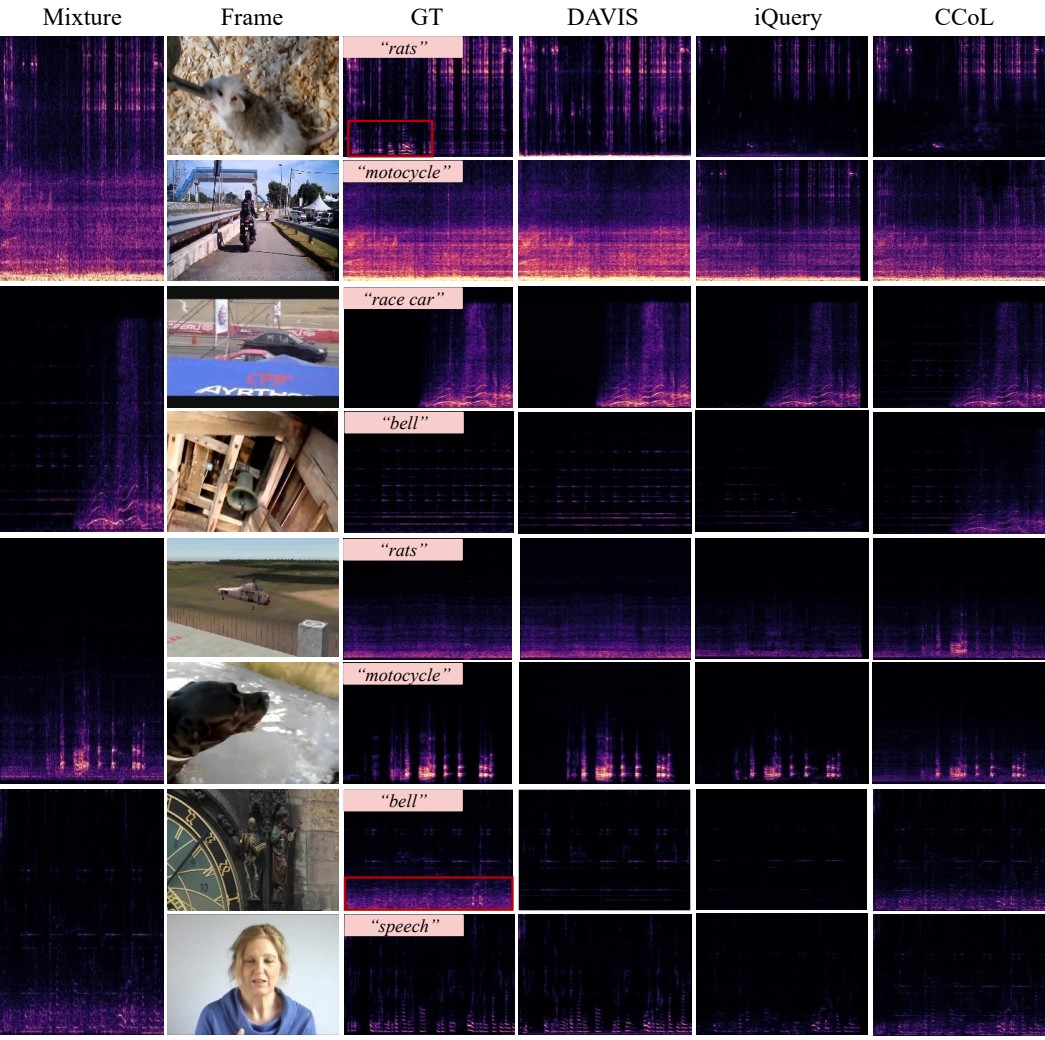

Figure 9: Visualizations of audio-visual separation results on the AVE dataset. Two sounds are mixed (mixture), and the referenced frame is provided to guide the separation. As the AVE dataset is unconstrained, there might be background noise existing in the videos. We mark the region that corresponds to the background noise in a red bounding box. We show the comparison between ground truth and DAVIS/iQuery/CCoL's predictions.

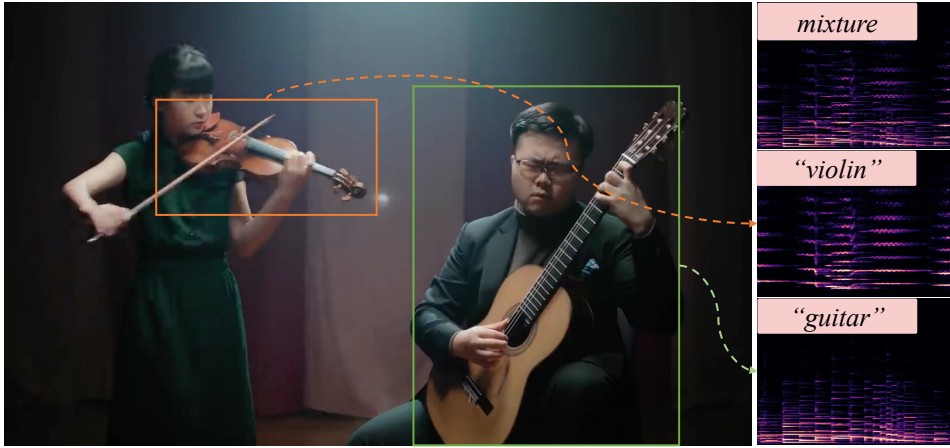

Figure 10: Real-world inference example. Different instruments in the bounding box are used as visual conditions. Video link: https://www.youtube.com/watch?v=Orp7IfOkTbg

## F  LIMITATION

Our proposed DAVIS framework incorporates the extraction of global visual embedding as a condition for visually-guided source separation. This technique, which utilizes global visual features, has been widely adopted in audio-visual learning (Zhao et al., 2018; Huang et al., 2023). Unlike methods that rely on pre-trained object detectors for extracting visual features, our framework does not have such a dependency. However, it may encounter limitations when trained on unconstrained video datasets. Intuitively, successful results can be achieved when the video contains a distinct sounding object, such as solo videos in the MUSIC dataset or videos capturing a sounding object performing a specific event in the AVE dataset. Nonetheless, this training assumption may not hold in more challenging scenarios, where multiple objects are likely producing sounds, rendering the global visual embedding inadequate for accurately describing the content of sounding objects. To address this issue, one possible approach is to adapt our framework to leverage more fine-grained visual features and jointly learn sounding object localization and visually-guided sound separation. This adaptation enables the model to utilize localized sounding object information to enhance the audio-visual association.

## G  FUTURE WORK

Our approach initiates the utilization of generative models for audio-visual scene understanding, paving the way for potential extensions to other multi-modal perception tasks like audio-visual object localization. Humans demonstrate the ability to imagine a "dog" upon hearing a "barking" sound, highlighting the potential of cross-modal generation in advancing audio-visual association learning. This implies that localization and separation tasks can be integrated into a single generative framework. In the future, we plan to explore the application of generative models to jointly address audio-visual localization and separation tasks.

