# OpenReview forum: "DAVIS: High-Quality Audio-Visual Separation with Generative Diffusion Models"
_ICLR.cc/2024/Conference — ICLR 2024 Conference Withdrawn Submission_

### Official Review · Reviewer_g6s8 · 2023-10-19

**Soundness:** 3 good
**Presentation:** 3 good
**Contribution:** 2 fair
**Rating:** 3
**Confidence:** 5

**Summary:**

In this work, the authors introduce a novel audio-visual speech separation framework grounded in a separation network and a diffusion process. They employ the existing UNet architecture with multiple attention mechanisms as their speech separation model. The exemplary performance of their model is demonstrated on both domain-specific and open-domain datasets, highlighting the advantages of the diffusion approach in the context of speech separation.

**Strengths:**

The paper is lucid and puts forth intriguing concepts. Employing a conditional diffusion model for generating separated sound sources marks a successful endeavor in the realm of auditory separation. The model achieves state-of-the-art separation performance on both the MUSIC and AVE datasets. Furthermore, the authors provide a detailed account, ensuring the training and inference of the model exhibit commendable reproducibility.

**Weaknesses:**

1. The architecture's resemblance to the application of the conditional diffusion model in the audio-visual speech separation domain — a method frequently seen in generative tasks within computer vision — seems more akin to method transfer, which dilutes the unique contribution of this paper. While the authors discuss the distinctions from existing conditional diffusion models in the appendix, these differences appear minimal. As I understood, the distinctions lie only in different inputs and outputs and variations in noise scheduling.

2. The discussion about existing diffusion-based speech separation models is perplexing. Different acoustic features as inputs don't inherently denote differences in diffusion model strategies. Furthermore, audio-visual speech separation and audio separation naturally embody different architectures and don't present direct comparability; they represent distinct tasks. For instance, would incorporating visual features into DiffSep yield results analogous to the DAVIS model to some extent?

3. Concerning dataset selection for comparisons, MUSIC and AVE exhibit similar characteristics, revolving around sound effects or musical events. Though the paper claims superior performance on open-domain datasets, I suggest the need to display model performance on widely used datasets in audio-visual separation, like LRS2, LRS3, and VoxCeleb2. These datasets, especially VoxCeleb2, with its thousands of unique speakers, might offer a more comprehensive view of open-domain results, given the limited types found in MUSIC and AVE.

4. The authors' assertion of speeding up the diffusion model's sampling steps might result from cleaner separation features guided by visual information. If so, the essentiality of the diffusion model itself becomes questionable.

5. Referring to Figure 4, I am skeptical about the over-separation caused by the DAVIS model. The removal of low-frequency parts on the right side of the Ground truth by the DAVIS model might not necessarily be due to capturing relevant semantic information. A single example isn't representative of the overall character. Thus, the authors' claim in Figure 4 that the DAVIS model can accurately learn audio-visual correlations and possesses the capability to capture complex data distributions doesn't seem well-founded.

**Questions:**

1. **Ablation Study**: I recommend including parameter count and computational cost in Table 2 for the ablation study section. This would help elucidate the reasons behind performance improvements when different components are introduced or altered. For example, substituting ResNet with a Time attention block in the 'Middle' column of Table 2 likely increases the model's parameters. Similarly, replacing ResNet with a Time-frequency attention block is expected to boost parameter count considerably. This leads to an essential question: is the observed performance enhancement attributed to the increase in the number of parameters?

2. **Sampling Steps in Diffusion**: I request the presentation of results for diffusion sampling steps in the range [1,5]. The underlying concern is whether even a single step could produce satisfactory results, raising a pertinent question about the necessity of the diffusion model in this specific task.

3. **Model Performance sans Diffusion**: To further address the doubt above, verifying the model's performance in the absence of the diffusion component is essential. Such an evaluation would illuminate the diffusion model's role and indispensability.

4. **Addressing Weakness (5)**: To counter the concern presented in the fifth Weakness, the authors should furnish statistical results from the complete test set. This would substantiate the validity of their claims and conclusions.

---

### Official Review · Reviewer_AwVq · 2023-10-30

**Soundness:** 3 good
**Presentation:** 4 excellent
**Contribution:** 3 good
**Rating:** 6
**Confidence:** 4

**Summary:**

This paper presents a diffusion model approach to the audio visual source separation problem. The method takes an audio mixture and an image of the desired source category and uses that to predict the output magnitude of the desired sound through an iterative diffusion denoising process. Experiments are conducted on several public datasets and comparison are made with existing methods.

**Strengths:**

This is a new take on the audio-visual source separation problem which incorporates recent advances in diffusion generative models. The method is well explained and easy to understand. Even though it seems like a combination of existing methods, there are a few sections such as fusing the visual features into the audio generator which are non-obvious and definitely novel. I also appreciate the detailed supplementary website provided. The authors walk through examples clearly which help the reader understand the performance and the comparison with other methods.

The evaluation is thorough, with two datasets compared using a variety of methods including ones from 2023.

**Weaknesses:**

I am not sure the authors need such a long recap of diffusion models in section 3.1

The contribution is limited as the authors are only synthesizing the magnitude spectrogram and using the phase of the original signal to invert the STFT. There are theoretical limitations to the SDR that can be achieved by this, and time domain models have beaten that limit. It would be a more substantial contribution if the authors operated on complex valued spectrograms. I don't think it would be that much more difficult to do this. The authors need to mention this limitation, and if they tried to use the complex spectrogram then they should describe challenges they faced when trying to operate in this domain.

I would like to see qualitative results on a real video, not just the artificial mix and separate examples in the supplementary webpage

**Questions:**

Did you try running the diffusion model on complex valued spectrograms?

Could models trained in this way generalize to more than 2 sources? Would they have to be re-trained specifically for a given number of sources?

---

### Official Review · Reviewer_gkhm · 2023-10-31

**Soundness:** 3 good
**Presentation:** 2 fair
**Contribution:** 2 fair
**Rating:** 3
**Confidence:** 4

**Summary:**

The paper proposes a diffusion-based audio-visual sound separation network named DAVIS.
Unlike existing methods that use mask-based separation methods, the paper aims to generate a more natural-sounding audio using a generative model.
The network is based on a separation u-net.
The use of u-net architecture for AVSS is popular, but this paper replaces mask regression with diffusion generative model.
The performance exceeds the baseline, which include some recent works.
The model is evaluated on two datasets of different domains: a musical instrument dataset and a more diverse general sounds dataset containing bell, kitchen sounds, dog barking, etc.

**Strengths:**

- The use of diffusion model in AVSS is novel to my knowledge, and is a reasonable method to improve performance on the task.
- The application of diffusion models to existing problems is popular and usually effective in most cases.

**Weaknesses:**

- Performance improvement against iQuery is mixed and unclear. iQuery requires class labels, but this is easy to obtain from the vision modality using a pre-trained classifier.
- Technical novelty is limited. The authors apply the popular diffusion model to AVSS. The addition of FIM is novel, but this only has a minor contribution to performance (Table 2 Left). I think this paper would be more suitable for audio/vision conferences in this respect.
- Inference time is probably much greater than existing works like iQuery, for a marginal improvement in performance. This should be compared and discussed.
- Fig 1 is unclear. The caption says t is passed to all modules whereas v is passed only to FIM, but this differentiation cannot be seen in the diagram. Also, you can't see that the abbreviation "FIM" (in the diagram) is "Feature Interaction Module" (in the caption) unless you look to Sec 3.3 several pages later.

**Questions:**

- Many recent works on using diffusion models for speech processing use two-stage process (e.g. Leng et al. 2205.14807, Popov et al. 2105.06337). Is there a reason for not considering or comparing to a two-stage method?
- The authors use temporal averaging in Visual Condition Aggregation. Could the authors provide reasons or comparisons for this design choice? Time-specific information could be useful for separation in many scenarios.

---

### Official Review · Reviewer_tFUD · 2023-11-01

**Soundness:** 2 fair
**Presentation:** 3 good
**Contribution:** 2 fair
**Rating:** 5
**Confidence:** 3

**Summary:**

This paper proposes a generative framework for visually-guided audio source separation task using diffusion model. The model architecture is specifically designed with the awareness of the time-frequency structure of audio and the interaction between audio and video. Experiments and audio samples demonstrate good separation result.

**Strengths:**

1. The proposed method achieves state-of-the-art performance on SDR and SIR on commonly used audio-visual source separation task.
2. The network architecture design is described in detail.

**Weaknesses:**

1. Weak contribution. Using diffusion model on source separation tasks is not novel [1,2,3]. Given these previous works, even if there is no existing diffusion-based visually guided source separation method, the insight brought by this paper is still very limited.
2. The relationship between audio-visual association and the performance of removing off-screen sound is very unclear. Firstly, if off-screen sound is removed, both SDR and SIR would drop. This will make these conventional metrics less effective. Secondly, there is only one sample showing the phenomenon of off-screen sound removal, which could be accidental. Thirdly, the paper did not offer a comparison of different audio-visual association choices on this effect. Actually, this effect can also be related to the generative nature of the proposed method. I suggest the author to evaluate more samples with human listener evaluation on this effect and compare the proposed method with at least one weak audio-visual association baseline.
3. The training dataset is relatively small compared to other diffusion models in image domain, which may cause overfitting. How did the authors alleviate the overfitting problem? How large is the proposed model?
4. Some details of this work are missing: The diffusion model generates amplitude spectrogram. How did the system convert it to audio waveform? Griffin-Lim method, or use pretrained vocoder, or other methods?

[1] Scheibler, Robin, et al. "Diffusion-based generative speech source separation." ICASSP 2023-2023 IEEE International Conference on Acoustics, Speech and Signal Processing (ICASSP). IEEE, 2023.

[2] Lu, Yen-Ju, et al. "Conditional diffusion probabilistic model for speech enhancement." ICASSP 2022-2022 IEEE International Conference on Acoustics, Speech and Signal Processing (ICASSP). IEEE, 2022.

[3] Serrà, Joan, et al. "Universal speech enhancement with score-based diffusion." arXiv preprint arXiv:2206.03065 (2022).

**Questions:**

See weakness.